# Learning Neural Exposure Fields for View Synthesis

**Michael Niemeyer**∗   **Fabian Manhardt**   **Marie-Julie Rakotosaona**   **Michael Oechsle**
**Christina Tsalicoglou**   **Keisuke Tateno**   **Jonathan T. Barron**   **Federico Tombari**
Google

## Abstract

Recent advances in neural scene representations have led to unprecedented quality in 3D reconstruction and view synthesis. Despite achieving high-quality results for common benchmarks with curated data, outputs often degrade for data that contain per image variations such as strong exposure changes, present, e.g., in most scenes with indoor and outdoor areas or rooms with windows. In this paper, we introduce Neural Exposure Fields (NExF), a novel technique for robustly reconstructing 3D scenes with high quality and 3D-consistent appearance from challenging real-world captures. In the core, we propose to learn a neural field predicting an optimal exposure value per 3D point, enabling us to optimize exposure along with the neural scene representation. While capture devices such as cameras select optimal exposure per image/pixel, we generalize this concept and perform optimization in 3D instead. This enables accurate view synthesis in high dynamic range scenarios, bypassing the need of post-processing steps or multi-exposure captures. Our contributions include a novel neural representation for exposure prediction, a system for joint optimization of the scene representation and the exposure field via a novel neural conditioning mechanism, and demonstrated superior performance on challenging real-world data. We find that our approach trains faster than prior works and produces state-of-the-art results on several benchmarks improving by over $55\%$ over best-performing baselines.

## 1   Introduction

Neural scene representations [8, 24, 33, 50] have revolutionized 3D vision due to their simple design, stable optimization, and state-of-the-art performance. As a result, they have now become the predominant representation for many 3D vision tasks ranging from 3D and 4D reconstruction [19, 30, 34–36, 53], to generative modeling [20, 29, 38, 44, 46], to view synthesis [3, 4, 26, 28].

While these methods have achieved unprecedented performance on existing view synthesis datasets, the majority of these benchmarks [3, 25] factor out several conditions that might occur in real world captures leading to a performance gap on such data. Among different phenomena, in particular strong exposure and appearance changes can lead to drastically degraded results (see Fig. 1a).

Prior works that consider exposure information during the optimization have explored this from a computer graphics perspective and usually aim to either reproduce images with certain exposures or recover a high dynamic range (HDR) representations that can be tonemapped via professional software [6, 13, 15, 47]. However, the goal of many applications is not only to reproduce input views at the same exposure, but to reconstruct scenes from RGB images in a 3D consistent manner and with an appearance that is most faithful to the real world. Another line of work does not consider input exposure and aim to recover a 3D consistent scene by explaining away per-image variations. For example, what has been widely adopted in practice is to use generative latent optimization (GLO) embeddings [5], e.g., as done in the pioneering work NeRF-W [21]. While being a robust solution

---

∗Corresponding author: `mniemeyer@google.com`

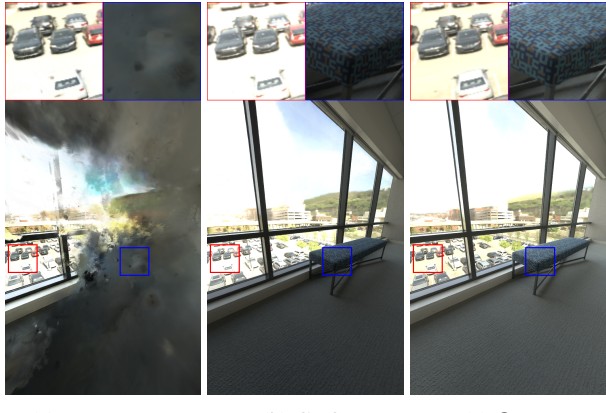

(a) Ignore Exp.     (b) GLO.     (c) **Ours**.

Figure 1: **Neural Exposure Fields.** While state-of-the-art neural fields [4] produce high-quality results on clean, well-curated datasets, the quality drops significantly for real-world captures if the exposure variation is ignored (1a). When equipped with per-view GLO embeddings [4, 21], the results improve (1b) but scene parts might be over- or underexposed. In contrast, our neural exposure field leads to high-quality 3D consistent scene appearance (1c) while no manual post-processing or reference view is required.

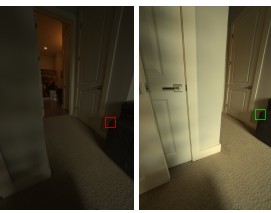

(a) Inconsistencies from 2D Tonemapping.

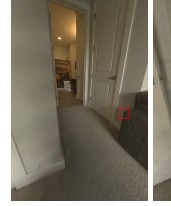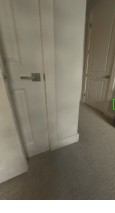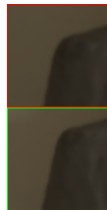

(b) 3D Consistent Appearance (Ours).

Figure 2: **3D Consistent Appearance.** Prior works [6, 15] use 2D tonemapping which can lead to inconsistent appearance of the same 3D point even in close-by views (2a). In contrast, our method produces 3D consistent appearance across the entire scene (2b).

for smaller appearance changes, we find that it leads to non-ideal color predictions when drastic variations are present, and parts of the scene can be over- or underexposed (see Fig. 1b). Recently, Bilarf [49] aims to learn more local bilateral mappings to explain per-image appearance changes. However a separate processing step is required together with a target appearance, e.g., given by an HDR image produced by an artist with professional software.

In this work, we present Neural Exposure Fields (NexF), a novel method to reconstruct scenes with high-quality appearance that shows well-exposed colors while being consistent in 3D (Fig. 1c). Our key insight is that we can learn a neural field that predicts an optimal exposure value per 3D point jointly with the scene representation. By aggregating information in 3D, as opposed to in 2D as done in capturing devices and previous works, our model is 3D consistent by design, freeing us from a separate per-image 2D tonemapping process (see Fig. 2). Further, we propose to learn neural exposure fields jointly with grid-based radiance fields using a novel latent exposure conditioning mechanism, leading to improved performance. In summary, **our contributions** are:

- A novel neural representation to predict optimal exposure values per 3D point.
- A system to jointly learn the exposure field and the neural 3D scene representation with a novel latent conditioning mechanism that produces high-quality view synthesis while being consistent in 3D.
- A thorough evaluation of the proposed system and baselines where we find that our approach improves over the state-of-the-art by more than $55\%$ in MSE.

We believe that our proposed system is a significant step towards bringing neural 3D scene representations closer to challenging real-world use cases while pushing the quality of view synthesis.

## 2 Related Work

**Neural Fields.** Since their emergence in 3D reconstruction [8, 24, 33, 50], neural fields have become a leading technique for various 3D vision tasks, e.g., 3D/4D reconstruction [19, 30, 34–36, 53], 3D generative modeling [7, 20, 29, 38, 44, 46], and view synthesis [3, 4, 26]. Their widespread adoption can be attributed to factors such as simplicity, strong performance, and stable optimization [50]. Unlike more traditional representations such as point clouds [37, 40, 41], voxels [22, 39], or meshes [12, 48], neural fields, especially when parameterized as an MLP, typically do not require

complex regularization, manually tuned initializations, or specialized optimization mechanisms. As a result, we chose to represent our exposure field as a neural field that is optimized end-to-end along with the neural scene representation.

**Neural Fields for View Synthesis.** In the context of view synthesis, Neural Radiance Fields (NeRF) [26] have produced unprecedented results by optimizing neural fields using volume rendering which demonstrated greater robustness compared to previous surface-based rendering methods [31, 45, 52]. This breakthrough inspired a series of follow-up works that improved, among others, the rendering quality [2–4] and speed [11, 18, 28, 32, 42, 43, 53]. Due to state-of-the-art performance, we use a modified version of ZipNeRF [4] as our scene representation. Note that we do not choose 3DGS [18] as scene representation in this work due to the degradation on challenging data and less stable joint optimization together with our neural exposure field parameterized by an MLP [32]. [2]

**Neural View Synthesis Beyond RGB Captures.** Several works have investigated how neural fields can be used for view synthesis from non-traditional captures. For example, [27, 47] investigate raw captures and [9, 54] low-light captures. Closer related to us, [6, 13, 15, 16, 47] investigate how tonemapping of HDR representations can be learned. HDRNeRF [15] proposes to condition a NeRF model on exposure by transforming the radiance to the log domain, achieving impressive results for HDR data. Nevertheless, HDRNeRF requires commercial 2D tonemapping software [14] and is also not 3D consistent. In [16], this model is extended with a field to predict local exposure. However, they require two training stages leading to excessive training times and rely on pseudo ground truth generated by the trained NeRF model to supervise the local exposure module, hence being restricted to synthetic, small-scale scenes. In contrast, our method is trained end-to-end by jointly optimizing the NeRF model and the neural exposure field. Further, our model produces high-quality, 3D consistent results even for challenging and large-scale captures with varying exposure.

**Neural View Synthesis for In-the-Wild Data.** While first view synthesis works were evaluated mostly on synthetic or very clean real-world captures, latter works shifted focus towards more in-the-wild data. The pioneering work NeRF-W [21] proposes to use Generative Latent Optimization (GLO) embeddings [5] to factor out per-image appearance variations. At test time, novel views are rendered with a fixed embedding (usually the zero vector) to generate a 3D consistent appearance. In [4], this approach is improved by optimizing an affine GLO transformation applied to the bottleneck of the MLP. While producing 3D consistent results, there is no direct control over the final predicted color and in the presence of strong exposure changes, it approaches a mean prediction which can produce over and underexposure for parts of the scene (see Fig. 1b). In Bilarf [49], per-view 3D bilateral grids are optimized and at test time, the appearance of a target image can be lifted to 3D in a second optimization stage. While producing good results, this is memory-heavy, time-consuming, and further requires a reference image that needs to be created by a professional artist or commercial HDR software [1]. In this work, we optimize a neural exposure field along with the scene representation that by design produces 3D consistent and high-quality, well-exposed colors without the need of an additional postprocessing step or a professionally-created reference image.

## 3 Method

We first describe our scene representation and latent exposure conditioning in Sec. 3.1. Next, we explain our neural exposure field along with the weighting criteria in Sec. 3.2. Finally, in Sec. 3.3 we outline the optimization strategy and implementation details. In Fig. 3, we show an overview over our method.

### 3.1 Neural Radiance Fields

A radiance field $f$ maps a point in 3D space $\mathbf{x} \in \mathbb{R}^3$ together with a viewing direction $\mathbf{d} \in \mathbb{S}^2$ to a volume density $\sigma \in \mathbb{R}^+$ and an RGB color value $\mathbf{c} \in \mathbb{R}^3$. The final color prediction for a pixel is approximated via quadrature using $N_s$ sample points along the ray [17]:

$$\mathbf{c}_{\text{pixel}} = \sum_{j=1}^{N_s} \tau_j \alpha_j \mathbf{c}_j \,, \quad \tau_j = \prod_{k=1}^{j-1}(1-\alpha_k) \,, \ \alpha_j = 1 - e^{-\sigma_j \delta_j} \,, \quad \delta_j = \|\mathbf{x}_{j+1} - \mathbf{x}_j\|_2$$

---

[2]Our model can directly be combined with distillation approaches such as [32].

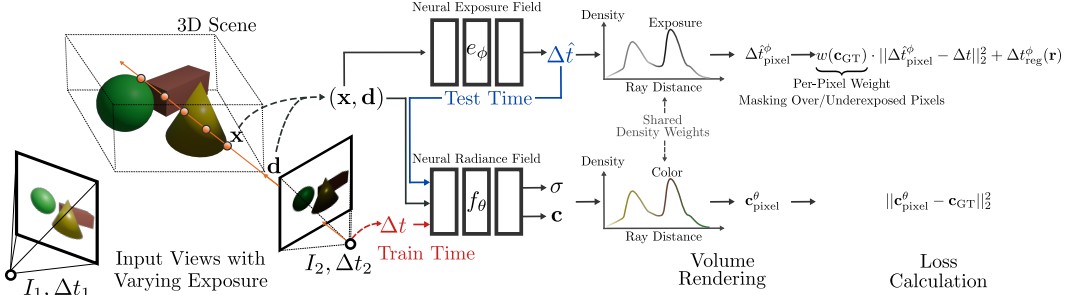

Figure 3: **Method Overview.** Our method takes as input a set of RGB images $\{I_i\}_i$ with exposure times $\{\Delta t_i\}_i$ and outputs a neural representation that produces high-quality, well-exposed appearance in a 3D consistent manner from arbitrary viewpoints. More specifically, during training, points are sampled along the ray and for each point $\mathbf{x}$, viewing direction $\mathbf{d}$, and input exposure $\Delta t$, the neural field $f_\theta$ predicts a density $\sigma$ and a color $\mathbf{c}$. The final color prediction $\mathbf{c}_{\text{pixel}}$ is obtained via volume rendering and the model is trained with the MSE loss on the input views with varying exposure. Similarly, the neural exposure field is trained by volume rendering the 3D predictions $\Delta \hat{t}$ to the image plane and backpropagating the reconstruction loss wrt. the input exposure weighted by how well the pixel is exposed and saturated. At test time, the neural field $f_\theta$ is instead conditioned on the neural exposure field predictions $\Delta \hat{t}$, producing high-quality, well-exposed novel views that are consistent in 3D where no 2D tonemapping nor target appearance produced by a professional is required.

where $\tau_j$ is transmittance, $\alpha_j$ is the alpha value for $\mathbf{x}_j$, and $\delta_j$ is the distance between neighboring samples.

**Parameterization.** A neural radiance field [26] optimizes an MLP $f_\theta$ parameterized by network weights $\theta$ using gradient descent with a reconstruction loss:

$$\mathcal{L}_f(\theta) = \sum_{\mathbf{r} \in \mathcal{R}_{\text{batch}}} \left\| \mathbf{c}_{\text{pixel}}^\theta(\mathbf{r}) - \mathbf{c}_{\text{GT}}(\mathbf{r}) \right\|_2^2 \tag{1}$$

where $\mathbf{r} \in \mathcal{R}_{\text{batch}}$ are sampled batches of rays. Using multisampling and a multi-resolution backbone [28], Zip-NeRF [4] produces state-of-the-art performance in view synthesis and we therefore adopt this architecture in our work. For complex real-world scenes like [51], we add affine GLO embedding vectors as proposed in [4] to increase robustness. Furthermore, we find that for forward-facing scenes, as present in the HDRNeRF [15] dataset, modifications for view-dependent color predictions are required to prevent overfitting. More specifically, for such scenes, we adjust the view-dependent branch from three layers with 256 hidden units to only two layers with 64 hidden units. We also reduce the bottleneck dimensionality to 15 and remove the skip connection to the second layer.

**Latent Exposure Conditioning.** Many real-world captures contain significant exposure variations and provide a per-image ground-truth exposure value. This lets us use per-image exposure as an input to our model alongside the RGB images, and condition the color prediction on exposure. We follow the classical nonparametric Conditional Random Fields (CRF) calibration [10, 15] and perform the conditioning in the logarithm radiance domain. Unlike prior works, however, we apply the transformation to the bottleneck vector of the MLP:

$$f_\theta(\mathbf{x}, \mathbf{d}, \Delta t(\mathbf{r})) = f_\theta^{\text{view}}(f_\theta^{\text{pos}}(\mathbf{x}) + \ln \Delta t(\mathbf{r}), \mathbf{d}), \tag{2}$$

where $\mathbf{x}$ is the sample point, $\mathbf{d}$ the viewing direction, $\Delta t(\mathbf{r})$ the exposure of corresponding ray $\mathbf{r}$, and $f_\theta^{\text{pos}}(\mathbf{x})$ the bottleneck vector. Crucially, we assume that $f_\theta^{\text{pos}}(\mathbf{x})$ predicts log radiance so that we do not require any transformation (see Fig. 5). We find that this latent conditioning leads to improved performance compared to prior works [15]. During training, we condition our NeRF $f_\theta$ on the input exposure to correctly reconstruct the input images following (1). However, at test time we condition on the neural exposure field prediction, which we will discuss in the following section.

### 3.2 Neural Exposure Fields

We propose to optimize a per-scene neural exposure field to predict the optimal exposure for each 3D point:

$$e_\phi : \mathbb{R}^3 \to \mathbb{R}, \, \mathbf{x} \mapsto \Delta \hat{t}(\mathbf{x}) \tag{3}$$

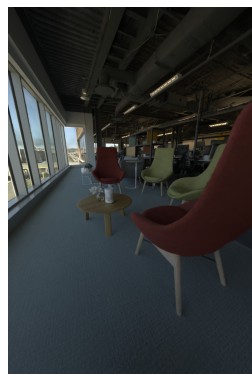
(a) RGB Prediction.

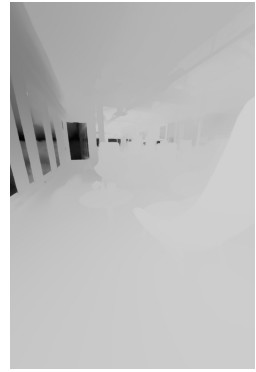
(b) Exposure Prediction.

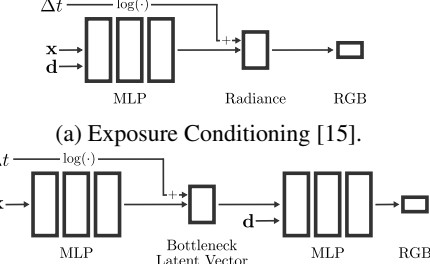
(a) Exposure Conditioning [15].

(b) Latent Exposure Conditioning (Ours).

Figure 4: **Exposure Visualization.** Instead of a single exposure per image as commonly done, we optimize a 3D neural exposure field predicting optimal exposure in 3D leading to well-exposed colors for all parts of the scene (see e.g. short exposure (dark) for outdoor and longer exposure (white) for darker indoor parts above).

Figure 5: **Exposure Conditioning.** While prior works [15] condition the color prediction on the log exposure by adding it onto the radiance (5a), we also perform the log transformation but condition in latent space instead (5b), leading to improved performance and more stable optimization.

where $\phi$ indicates the network parameters and $\Delta \hat{t}(\mathbf{x})$ the predicted exposure at 3D point $\mathbf{x}$. It is important to note that, in contrast to the conventional formulation of exposure, we define exposure not as a function of the camera location and viewing direction, but as a function of 3D position. This is based on the hypothesis that there exists an optimal exposure value for each 3D point for which the predicted color is well-exposed (see Fig. 4). This lets us learn high-quality appearances for high-dynamic range scenarios that are consistent in 3D, in contrast to prior works [6, 15] that perform tonemapping on the 2D image plane leading to inconsistent synthesis across views.

**Parameterization.** We parameterize our neural exposure field $e$ as a fully-connected MLP. We optimize this neural exposure field alongside our scene representation $f_\theta$. As we aggregate exposure information in 3D, in contrast to prior works and capturing devices that operate in 2D, we can define criteria to obtain the best-possible "ideal" exposure per point which we will do in the following.

**Objective 1: Well-Exposedness.** A pixel's color is often referred to as over- or underexposed if the color value is close or at the clipping boundaries. More specifically, we measure well-exposedness as

$$w_{\text{exp}}(\mathbf{c}) = \prod_i \exp\left(-\frac{(c_i - 1/2)^2}{\sigma_{\text{exp}}}\right), \tag{4}$$

where $\sigma_{\text{exp}}$ is a hyperparameter controlling the sharpness of the weight curve.

**Objective 2: Saturation.** While our main focus lies on learning ideal exposure per 3D point, we draw inspiration from classical image processing [23] and further integrate an objective that aims to obtain well saturated colors. More specifically, we define

$$w_{\text{sat}}(\mathbf{c}) = \sqrt{\frac{1}{3}\sum_i (c_i - \boldsymbol{\mu}_c)^2}, \tag{5}$$

where $\boldsymbol{\mu}_c$ defines the mean of the R, G, and B value of $\mathbf{c}$.

**Regularization.** Given that there is no target ground truth 3D exposure available, we find that adding regularization in 3D space leads to better results. Intuitively, we hypothesize that the ideal exposure changes smoothly in 3D space as close-by 3D points are presumably well-exposed at similar exposure times. More specifically, we calculate the squared distance between predicted exposure of nearby 3D points

$$\Delta t_{\text{diff}} = \|e_\phi(\mathbf{x}) - e_\phi(\mathbf{x} + \epsilon)\|_2^2, \tag{6}$$

where $\epsilon \sim \mathcal{N}(0, 0.05)$ is a small 3D normally-distributed noise vector. As outlined next, we penalize $\Delta t_{\text{diff}}$ to regularize our neural exposure field that in turn leads to improved performance.

## 3.3   Optimization

We do not assume to have a target RGB or HDR appearance but rather to learn optimal exposure from 2D observations to produce high-quality, 3D consistent appearance. Our key idea is to selectively backpropagate the exposure of input views only if the color is well exposed and well saturated, according to our objective functions. This way, we do not need to assume that every pixel is well exposed in certain views which is often not the case, but rather only need to make the assumption that most areas in the 3D scene show well exposed colors in *some* view. More specifically, we observe that we can render predicted exposure similarly to color as $\Delta t_{\text{pixel}} = \sum_{j=1}^{N_s} \tau_j \alpha_j \Delta \hat{t}_j$ where $\tau$, $\alpha$, and $\delta$ are the same as in (3.1). Next, we define a per-pixel weight

$$w(\mathbf{c}) = w_{\text{exp}}(\mathbf{c})^{\lambda_{\text{exp}}} \cdot w_{\text{sat}}(\mathbf{c})^{\lambda_{\text{sat}}}, \tag{7}$$

where $\lambda_{\text{exp}}$ and $\lambda_{\text{sat}}$ are hyperparameters controlling the influence of the respective mask. For regularization, we can similarly render the exposure difference of nearby sample points $\Delta t_{\text{reg}} = \sum_{j=1}^{N_s} \tau_j \alpha_j \Delta t_{\text{diff}}^j$. We train our neural exposure field $e_\phi$ with the input exposure

$$\mathcal{L}_e(\phi) = \sum_{\mathbf{r} \in \mathcal{R}_{\text{batch}}} w(\mathbf{c}(\mathbf{r})) \cdot \left\| \Delta \hat{t}_{\text{pixel}}^\phi(\mathbf{r}) - \Delta t(\mathbf{r}) \right\|_2^2 + \Delta t_{\text{reg}}^\phi(\mathbf{r})$$

**Joint Optimization.** We train our neural scene representation $f_\theta$ and exposure field $e_\phi$ end-to-end from the input captures and the full loss formulation becomes

$$\mathcal{L}(\theta, \phi) = \mathcal{L}_f(\theta) + \mathcal{L}_e(\phi). \tag{8}$$

It is important to note that we only require RGB images with exposure as input and no additional labels are required. Using our color-based criteria, we produce high-quality 3D consistent view synthesis results in high dynamic range scenarios while prior works require, e.g., a target HDR image from an artist or access to professional HDR software.

**Implementation Details.** We parameterize our neural exposure field as a fully-connected MLP with ReLU activation and four hidden layers with a hidden dimension of 128. For rendering the predicted exposure to the image plane, we re-use the alpha blending weights calculated for the RGB rendering pass (see (3.1)) for faster performance and detach the gradients so that the scene geometry prediction is not affected by the exposure field prediction branch. In the NeRF model, we use a bottleneck vector dimension of 256 per default except for forward-facing scenes where we use a dimension of 15. We train both models jointly for $10,000$ iterations ( approx. 10 min.) on forward-facing and for $25,000$ iterations (approx. 30 min.) on room- and apartment-sized scenes on 8 V100 GPUs. For additional details on the NeRF parameterization, we refer the reader to [4]. We set the weighting-related hyperparameters as $\sigma_{\text{exp}} = 0.05$, $\lambda_{\text{exp}} = 0.1$, and $\lambda_{\text{sat}} = 1$.

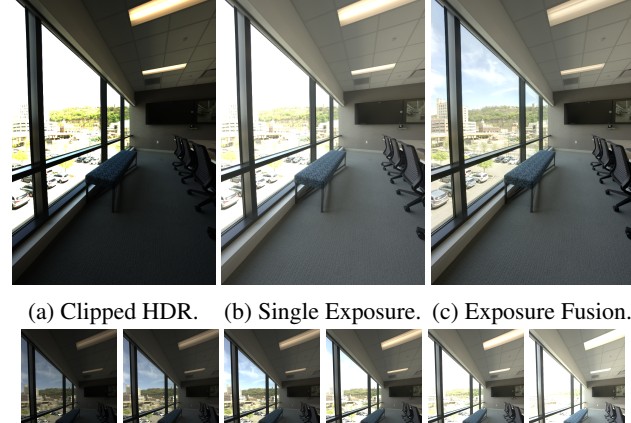

(a) Clipped HDR.  (b) Single Exposure.  (c) Exposure Fusion.

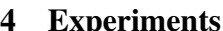

(d) Varying Exposure Values.

Figure 6: **Exposure Fusion.** During training, each input image is observed with only a single randomly-sampled exposure (6d). For evaluation, we apply exposure fusion (6a) to obtain higher-quality and well-exposed target images (6c) compared to the default single exposure images (6b).

## 4   Experiments

**Data.** To evaluate our neural scene representation with the latent exposure conditioning, we report reconstruction results on the HDRNeRF [15] dataset (MIT). To further evaluate our model on real-world and scene-level captures,

we also report results on the recently proposed Eyeful Tower dataset v2 [51] (CC BY-NC I 4.0). In contrast to other common view synthesis benchmarks like [3], the Eyeful Tower dataset contains HDR images that we can use to simulate varying exposure and to produce meaningful target images. Finally, we test our model on in-the-wild phone captures from [49] (Apache 2.0) and a large-scale scene from [4] (CC-BY).

**Metrics.** For the exposure reconstruction experiments, we follow prior works [6, 15] and report view synthesis metrics PSNR, SSIM, and LPIPS against the ground truth images for both in-distribution and out-of-distribution exposure values. For evaluating our neural exposure field on room-level scenes, we use the HDR data from the Eyeful Tower [51] dataset that allows us to generate tonemapped images with different exposures. For the training set, for each view we sample an exposure value from an exposure set $(\frac{1}{16}, \frac{1}{8}, \frac{1}{4}, \frac{1}{2}, 1, 2)$ so that every pose is only observed once with one exposure. For evaluation, we use the HDR data to generate each test view with all exposures from our exposure set and then produce the target image by applying classical exposure fusion [23] as illustrated in Fig. 6. To measure how well our method and baselines produce well-exposed reconstructions, we report PSNR, SSIM, LPIPS against these target images on the test set.

**Baselines.** For the exposure reconstruction experiments, we compare against state-of-the-art methods NeRF [26], ZipNeRF [4], 3DGS [18], NeRF-W [21], HDRNeRF [15], and HDR-GS [6], where HDR-GS is the only method that further requires the HDR captures as input, while all other methods only use RGB at sampled exposure. For the results on the Eyeful Tower scenes, we report our method and all baselines with the same ZipNeRF [4] backbone to enable a fair comparison. For these experiments, we compare against ZipNeRF a.) without any modification to handle varying exposure ("Ignore Exposure"), b.) with GLO embeddings [5, 21], c.) with affine GLO embeddings [4], d.) HDRNeRF*[15] for which we train the NeRF model with exposure input and at test time, we render with constant mean exposure over all input images.

## 4.1 View Synthesis with Input Exposure

In the first experiment, we evaluate the performance of our chosen neural scene representation and the novel latent exposure conditioning mechanism. The task is to reconstruct test views at different input exposures.

Table 1: **Comparison on HDRNeRF.** We find that for both in-distribution (IN) and out-of-distribution (OOD) exposure values, our method overall performs best while training significantly faster than prior works. *Note that HDR-GS [6] uses the HDR data during training, while all other methods only rely on RGB images at sampled exposure.

|  | Time | ID Exposure $(t_1, t_3, t_5)$ | | | OOD Exposure $(t_2, t_4)$ | | |
|---|---|---|---|---|---|---|---|
|  | min. | PSNR ↑ | SSIM ↑ | LPIPS ↓ | PSNR ↑ | SSIM ↑ | LPIPS ↓ |
| NeRF [26] | 405 | 13.97 | 0.555 | 0.376 | 14.51 | 0.522 | 0.428 |
| ZipNeRF [4] | 11 | 19.00 | 0.682 | 0.142 | 19.73 | 0.724 | 0.125 |
| 3DGS [18] | 38 | 19.46 | 0.690 | 0.276 | 18.97 | 0.778 | 0.309 |
| NeRF-W [21] | 437 | 29.83 | 0.936 | 0.047 | 29.22 | 0.927 | 0.050 |
| HDRNeRF [15] | 542 | 39.07 | 0.973 | 0.026 | 37.53 | 0.966 | 0.024 |
| HDR-GS* [6] | 34 | 41.10 | 0.982 | 0.011 | 36.33 | 0.977 | 0.016 |
| **Ours** | 11 | 42.54 | 0.988 | 0.014 | 38.36 | 0.984 | 0.021 |

**Quantitative Results.** In Tab. 1 we observe that our method overall leads to the best performance while training $3\times$ faster than the fastest baselines. Compared to HDRNeRF [15], the prior state-of-the-art that also uses only RGB sampled at exposure values during training, we improve by 3.5 PSNR from 39.07 to 42.54 on the in-distribution (ID) exposure values and by 0.8 PSNR from 37.53 to 38.36 on the out-of-distribution (OOD) exposure values. Compared to HDR-GS [6] that requires not only sampled RGB but the full HDR capture during optimization, we also improve in PSNR (by 1.4 on ID and 2.0 on OOD exposure) and SSIM while achieving a slightly higher LPIPS value. It is crucial to note that RGB input sampled at different exposures is a common scenario for real-world captures, while HDR captures require a professional setup and usually are only available for professionally captured or synthetic environments. We conclude that our method performs best for view synthesis with input exposure compared to SOTA baselines while training at least $3\times$ faster.

Table 2: **Ablation on HDRNeRF.** We ablate our proposed view-dependent MLP architecture for forward-facing scenes (w/o Our View. MLP) and the latent exposure conditioning (w/o Lat. Con.). Our full model leads to the best results on all metrics.

| | ID Exposure $(t_1, t_3, t_5)$ | | | OOD Exposure $(t_2, t_4)$ | | |
| --- | --- | --- | --- | --- | --- | --- |
| | PSNR ↑ | SSIM ↑ | LPIPS ↓ | PSNR ↑ | SSIM ↑ | LPIPS ↓ |
| w/o Our View. MLP | 33.85 | 0.928 | 0.104 | 28.20 | 0.880 | 0.170 |
| w/o Lat. Con. | 39.88 | 0.979 | 0.038 | 38.33 | 0.978 | 0.034 |
| **Ours** | 42.54 | 0.988 | 0.014 | 38.36 | 0.984 | 0.021 |

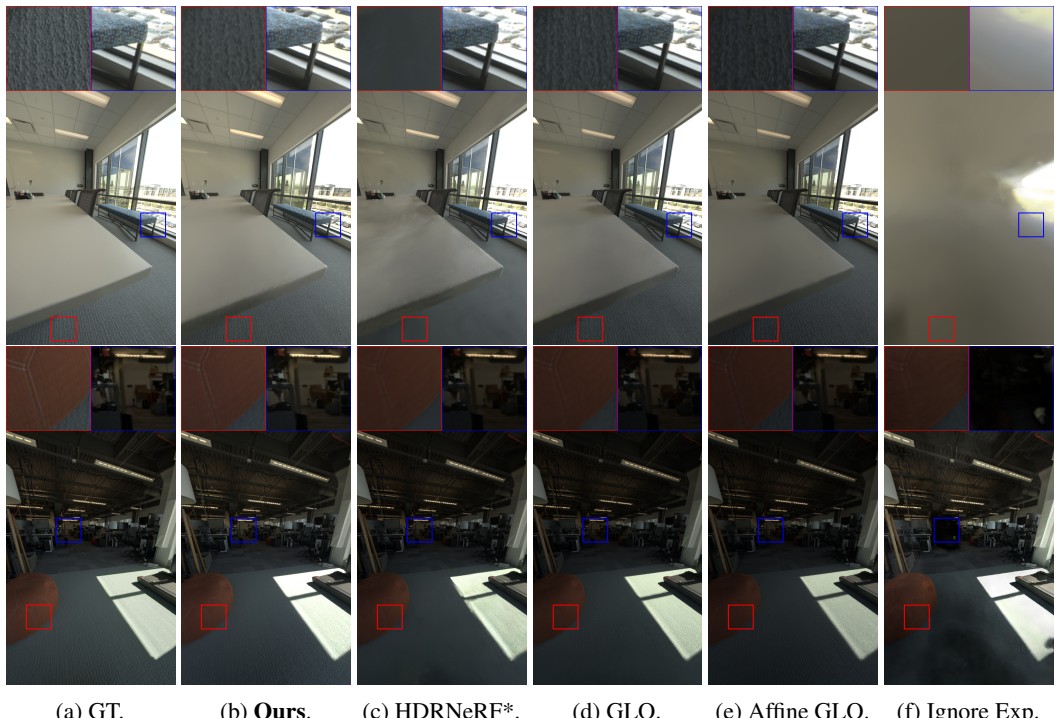

(a) GT.  (b) **Ours**.  (c) HDRNeRF*.  (d) GLO.  (e) Affine GLO.  (f) Ignore Exp.

Figure 7: **Qualitative Results on Eyeful Tower.** Results from our model and baselines for the `office_view2` and `riverview` scenes.

**Ablation.** We additionally perform an ablation study in Tab. 2 for the exposure reconstruction experiment to verify the relevance of important components. We find that our model without our view-dependent MLP parameterization (see Sec. 3.1) leads to degraded results and tends to overfit to input images. If we do not use our latent exposure conditioning in our model, we find that the quality drops as the model is less expressive and predicted views are less sharp. Our full model leads to the best results on all metrics and we conclude that both contributions are required for high performance.

### 4.2 View Synthesis with Neural Exposure Field

Table 3: **Comparison on Eyeful Tower.** Also quantitatively, we find that our method performs best compared to state-of-the-art baselines that all share the same ZipN-eRF [4] backbone.

| | PSNR ↑ | SSIM ↑ | LPIPS ↓ |
| --- | --- | --- | --- |
| Ignore Exposure | 16.72 | 0.682 | 0.444 |
| Affine GLO [4] | 20.13 | 0.815 | 0.263 |
| GLO [21] | 21.20 | 0.824 | 0.298 |
| HDRNeRF* [15] | 22.82 | 0.836 | 0.311 |
| **Ours** | 26.48 | 0.876 | 0.234 |

Table 4: **Ablation on Eyeful Tower.** We ablate our two criteria for the exposure field optimization (w/o Well-Exposedness and w/o Saturation), our regularization strategy (w/o Regularization), and the affine GLO embeddings (w/o A-GLO).

| | PSNR ↑ | SSIM ↑ | LPIPS ↓ |
| --- | --- | --- | --- |
| w/o Well-Exposedness | 25.92 | 0.868 | 0.247 |
| w/o Saturation | 24.30 | 0.867 | 0.241 |
| w/o Regularization | 24.84 | 0.856 | 0.258 |
| w/o A-GLO | 27.44 | 0.854 | 0.306 |
| **Ours** | 26.48 | 0.876 | 0.234 |

Next, we evaluate how well our method performs at view synthesis with 3D consistent appearance on room- and apartment-sized scenes with our neural exposure field. In contrast to before, the task is not to reproduce the appearance at an input exposure, but rather to learn consistent and well-exposed appearance across the entire scene.

**Quantitative Results.** We find that our method performs best compared to baselines in all metrics (see Tab. 3). In PSNR and SSIM, HDRNeRF* [15] performs second best and we improve over it significantly by 57% in MSE (+3.7 PSNR, +0.04 SSIM). In LPIPS, affine GLO [3] performs second best and we improve over it by 12% (−0.03 LPIPS).

**Qualitative Results.** Qualitatively, we observe a similar trend (see Fig. 7). In particular, ignoring exposure leads to floating artifacts and severe scene degradation. Further, all remaining baselines exhibit over- and underexposure in various parts of the scene and lose fine details. In contrast, our method produces well-exposed colors for the entire scene including fine details and far away regions.

**Ablation.** In Tab. 4, we ablate the main components of our method. We find that both criteria, well-exposedness as well as saturation, are required to achieve high-quality results. Without the well-exposedness criterion, especially the LPIPS metric drops, proving that well-exposed colors are perceptually very important. On the other hand, without the saturation criterion, the PSNR metric drops, demonstrating its relevance for faithfully reconstructing well-fused colors (GT). If we remove our proposed regularization, the neural exposure field predictions do not generalize as well to novel views, causing a performance drop.

Finally, without the affine GLO embeddings, we find that while PSNR slightly increases, both SSIM and LPIPS worsen significantly due to overall more blurry results. Ultimately, we conclude that all components of our method are crucial to achieve the best performance.

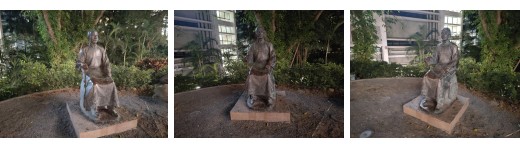

(a) GT Test Images.

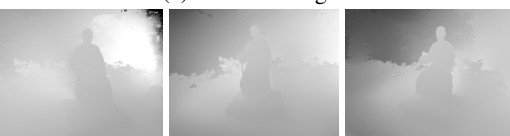

(b) Our Exposure Field Predictions.

**In-the-Wild Captures.** Finally in Fig. 8, we additionally show qualitative results for the in-the-wild phone capture `statue` from [49]. We observe that while the input (both train and test) images exhibit strong exposure changes, our method is able to produce a 3D consistent exposure that in turn leads to visually appealing, 3D consistent view synthesis. Crucially, compared to prior works, our method does not require exposure as input at test time nor a reference target appearance created by an artist with professional software.

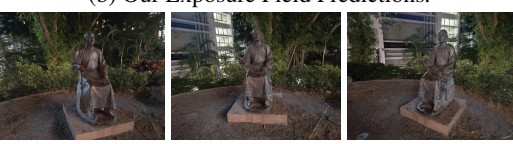

(c) Our RGB Predictions with 3D Consistent Exposure.

Figure 8: **In-the-Wild Captures.** For in-the-wild captures [49], both training and test images exhibit strong exposure changes (8a). Our method learns a 3D consistent exposure field (8b) resulting in 3D consistent high-quality view synthesis (8c).

**Large-Scale Reconstruction.** In Fig. 9 and 10, we additionally report qualitative results on `Alameda`, a house-sized scene with varying input exposure, from the ZipNeRF dataset [4]. For all methods, we use the same ZipNeRF [4] backbone. We find that ignoring the input exposure leads to degenerate view synthesis. Optimizing affine GLO embeddings [4, 21] improves the overall quality but the renderings can still suffer from over- and underexposure. Our method, in contrast, produces 3D consistent and well-exposed colors for all parts of the scene, while not requiring a complex multi-exposure capture as input. It is wort noting that in Fig. 9 and 10, our method improves over the ground truth view that was captured with a single exposure value.

**3D Exposure.** While in the traditional sense, exposure is a camera-dependent property, our method's "optimal exposure" is a conceptual departure acting as a spatially-varying variable. It is not directly a physical property but rather a learned representation that guides our model to produce high-quality, well-exposed images, similar to how a photographer might locally adjust exposure in high dynamic range photography. By learning a 3D exposure field, our model finds a per-point exposure value that prevents clipping and under or over-saturation for each point.

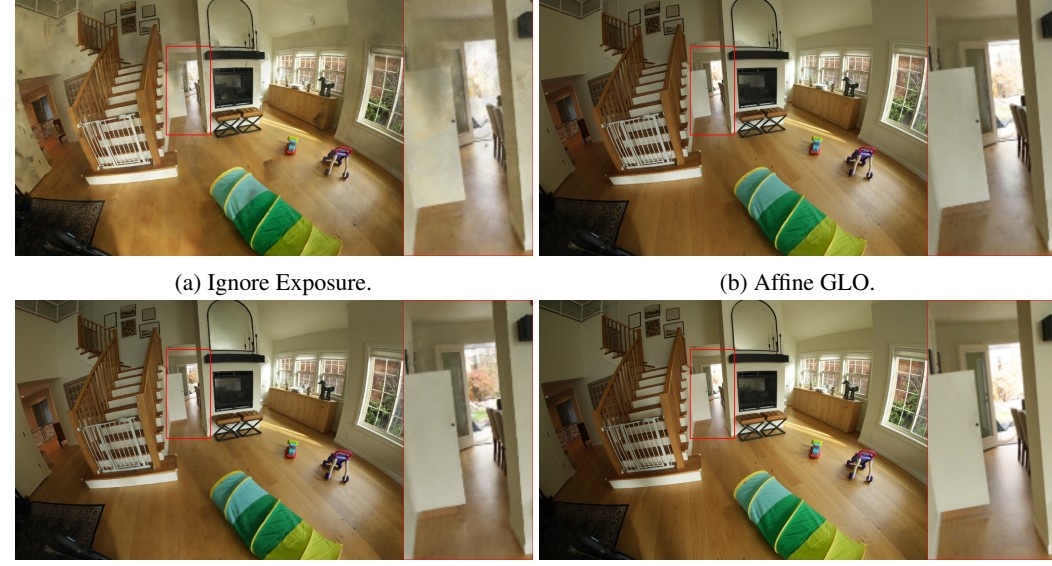

(a) Ignore Exposure.

(b) Affine GLO.

(c) **Ours**.

(d) Ground Truth.

Figure 9: **Large-Scale View Synthesis.** While ignoring the input exposure (9a) leads to degenerate view synthesis, optimizing affine GLO embeddings [4, 21] (9b) improves results but output images still suffer from over- and underexposure. In contrast, our method (9c) achieves 3D consistent and well-exposed color for the entire large-scale scene without requiring a complex multi-exposure capture, and even improves over the GT test view that was captured with a single exposure (9d).



(a) GT Test View.

(b) Our RGB Prediction.

(c) Our Exposure Field Prediction.

Figure 10: **High-Dynamic Range.** While input train and test views might show over- or underexposure (10a) as no complex multi-exposure capture was performed, our model achieves well-exposed colors (10b) for the entire scene thanks to the predictions of the 3D neural exposure field (10c).

**Limitations.** While our method produces state-of-the-art view synthesis for scenes with varying exposure, results might degrade for extreme lighting conditions such as very low-light or extreme over-exposed captures as well as for complex lighting effects such as semi-transparency and strong reflections.

## 5    Conclusion

We presented Neural Exposure Fields (NExF), a novel method for view synthesis that produces high-quality 3D-consistent appearances for high dynamic range scenarios. The key idea is to learn a 3D neural exposure field from 2D observations by considering a pixel's well-exposedness and saturation for backpropagation, and to train it jointly with our neural scene representation with a novel latent exposure conditioning. In contrast to previous works, our method does not require HDR captures nor target appearances produced by professional artists and costly software. We find that our method produces high-quality scene reconstructions with well-exposed colors for all parts of the scene, even in challenging high dynamic range scenarios such as indoor rooms with large window fronts overlooking outdoor areas. Also quantitatively, we find that our approach leads to state-of-the-art performance improving by more than 55% over the best-performing baseline.

**Acknowledgments.** We would like to thank Peter Zhizhin, Peter Hedman, and Daniel Duckworth for fruitful discussions and advice, and Cengiz Oztireli for reviewing the draft.

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
