# OpenReview forum: "Learning Neural Exposure Fields for View Synthesis"
_NeurIPS.cc/2025/Conference — NeurIPS 2025 poster_

### Official Review · Reviewer_wXZ9 · 2025-06-10

**Clarity:** 3
**Significance:** 3
**Originality:** 3
**Rating:** 5
**Confidence:** 4

**Summary:**

This paper proposes Neural Exposure Fields (NExF), a NeRF-based framework that robustly handles exposure-varying input images by learning the exposure and scene jointly. Specifically, the key designs of the proposed method include (1) the latent exposure conditioning that conditions the bottleneck latent vector (rather than output radiance) on the log exposure; (2) a neural exposure field that encodes exposure in the 3D space, and a carefully designed loss function suite for optimization. The results demonstrate superior performance compared to baselines.

**Questions:**

# Major Questions
1. Please provide a reasonable explanation of the design of latent exposure conditioning. An ablation study comparing latent exposure conditioning and explicit exposure conditioning can further validate the method.
2. Please provide a reasonable explanation of the design of 3D exposure modeling. An ablation study is also preferred.
3. Please provide comparisons with the Bilarf baseline.

# Minor Questions
1. Positional encoding of the neural fields: Does the paper use Instant-NGP hashgrid as the positional encoding (as in Zip-NeRF), or the sinusoidal positional encoding? This implementation detail should be mentioned.
2. Ablation study: I don't quite understand the "w/o view MLP" ablation in Tab. 2. Isn't the view-dependent MLP a standard design in all kinds of NeRF works? Is there anything special in your design?

**Ethical Concerns:**

["NO or VERY MINOR ethics concerns only"]

**Final Justification:**

My questions and concerns are mostly resolved, so I decided to raise the score to 5.

**Limitations:**

yes

**Quality:**

3

**Strengths And Weaknesses:**

# Strengths
1. Good paper writing: The paper is generally well-written and easy to follow. I appreciate the author's efforts in illustrating key design components (Figs. 4 and 5), which greatly improve readability.
2. Comprehensive design and good results: The proposed framework is carefully and reasonably designed. The multi-objective loss function for the neural exposure field comprehensively considers different factors that may affect exposure and is validated by the ablation studies. Overall, this comprehensively designed framework yields a superior result compared to the baselines.

# Weaknesses
1. Explanations of latent exposure conditioning. Although the authors claimed that their proposed latent exposure conditioning is more effective than the previous work's exposure conditioning, the authors did not provide sufficient explanations and evidence for it. The bottleneck latent vector lies in the latent space without explicit physical meaning, so directly adding a physical parameter (the log exposure) to it does not make a lot of sense, compared to previous work's explicit exposure conditioning, which is more interpretable.
2. Explanations of 3D exposure modeling. This proposed framework encodes exposure in the 3D space, which is also weird. In my opinion, exposure is a camera-dependent property, rather than space-dependent, since it is associated with the camera image processing. For a single image, the camera exposure on all pixels should be consistent, but modeling exposure in 3D may cause inconsistency.
3. Missing baseline: The paper has cited the recent work Bilarf (and also evaluates on its dataset), but does not compare with Bilarf.

---

> ### Author Rebuttal · Authors · 2025-07-30
>
> Thanks for the detailed review. We are encouraged that you find our paper "well-written and easy to follow" and our method to be "carefully and reasonably designed" with "superior result[s] compared to the baselines". We address concerns in the following:
>
> > Weakness: Explanations of latent exposure conditioning. Although the authors claimed that their proposed latent exposure conditioning is more effective than the previous work's exposure conditioning, the authors did not provide sufficient explanations and evidence for it. The bottleneck latent vector lies in the latent space without explicit physical meaning, so directly adding a physical parameter (the log exposure) to it does not make a lot of sense, compared to previous work's explicit exposure conditioning, which is more interpretable.
>
> > Corresponding Question: Please provide a reasonable explanation of the design of latent exposure conditioning. An ablation study comparing latent exposure conditioning and explicit exposure conditioning can further validate the method.
>
> We appreciate the comment and agree that operating in a latent space instead of RGB color directly adds a layer of abstraction. This design choice leads to a significant performance improvement of 45% in MSE (+2.7dB in PSNR), as can be seen in Table 2 of the main paper. We therefore believe that it is still a valuable contribution and design choice. To investigate additional potential design choices, we ran the following ablation study on the HDRNeRF Dog scene:
>
> | Method | PSNR  | SSIM  | LPIPS  |
> |---|---|---|---|
> | Ours | 41.25 | 0.993   | 0.008  |
> | Ours without Latent but with Explicit Exposure Conditioning  | 40.14 | 0.989  | 0.020  |
> | Ours without Exposure Log Transform  | 34.18 | 0.965  | 0.021  |
> | Ours without Exposure Conditioning  | 18.38 | 0.753  | 0.103  |
>
> We find that all our design choices lead to improved performance. We added this to the final version. Further, we note that operating in latent space have become common in modern computer vision models and has led to breakthroughs in related subfields (see e.g. latent diffusion in generative modeling (LDMs) or positional encoding in view synthesis (NeRF)).
>
>
> > Weakness: Explanations of 3D exposure modeling. This proposed framework encodes exposure in the 3D space, which is also weird. In my opinion, exposure is a camera-dependent property, rather than space-dependent, since it is associated with the camera image processing. For a single image, the camera exposure on all pixels should be consistent, but modeling exposure in 3D may cause inconsistency.
>
> > Corresponding Question: Please provide a reasonable explanation of the design of 3D exposure modeling. An ablation study is also preferred.
>
> Thanks for your comment. We agree that exposure itself is view-dependent as it is a scalar that is defined per image and hence camera pose. In this work, our goal is not to reproduce input exposure, but instead learn an **optimal exposure value** per 3D point. This enables, as can be seen in the fly-through videos in the supp. mat., smooth and stable color prediction across the entire scene, while per image/camera pose exposures can vary drastically as shown in Figure 2 of the main paper.
> Further, below we report an additional ablation study comparing our default method to our method with the modification of adding the viewing direction onto the bottleneck vector, producing similar results:
>
> | Method | PSNR  | SSIM  | LPIPS  |
> |---|---|---|---|
> | Ours | 41.25 | 0.993   | 0.008  |
> | Ours + View. Dep.  | 41.33 | 0.993  | 0.008  |
>
>
> > Weakness: Missing baseline: The paper has cited the recent work Bilarf (and also evaluates on its dataset), but does not compare with Bilarf.
>
> Thanks for the comment. Please note that in our benchmarks, we focus on reporting results for state-of-the-art methods that are comparable to our model. For example, Bilarf, as discussed in the main paper (see L.99 - L.101), requires, in contrast to our method and baselines, a target reference image produced with HDR software as input, and hence is not directly comparable. To test the performance of our method on in-the-wild data (without any target reference image as input), we report a qualitative result on the Bilarf dataset in Figure 8 of the main paper.
> Despite not being directly comparable in terms of input (see above), we nevertheless ran our method on the Statue scene of the Bilarf dataset producing the following comparison:
>
> | Method | PSNR  | SSIM  | LPIPS  |
> |---|---|---|---|
> | Bilarf | 23.33 | 0.826   | 0.137  |
> | Ours  | 24.23 | 0.828  | 0.152  |
>
> We find that our method overall performs slightly better than Bilarf on its own benchmark. Further, it is worth noting that Bilarf optimizes per-image 3D grids and requires two stage optimization, leading to significant memory requirements for larger captures and longer optimization times. We added this extended discussion to the final version.
>
>
> > Minor question: Positional encoding of the neural fields: Does the paper use Instant-NGP hashgrid as the positional encoding (as in Zip-NeRF), or the sinusoidal positional encoding? This implementation detail should be mentioned.
>
> Thanks. We use a multi-resolution hash grid backbone similar to Instant-NGP as positional encoding. We clarified this in the final version.
>
> > Minor question: Ablation study: I don't quite understand the "w/o view MLP" ablation in Tab. 2. Isn't the view-dependent MLP a standard design in all kinds of NeRF works? Is there anything special in your design?
>
> Thanks for the comment. The method name reads "w/o Our View. MLP" not "w/o View. MLP", i.e. you missed the "Our" of the description. In the referenced Table 2 of the main paper, we compare our proposed view-dependent MLP architecture to the view-dependent architecture proposed in ZipNeRF, finding that our proposed architecture leads to improved performance. For more details on this, please see L.121 - L.125 of the main paper.

---

> > ### Comment · Reviewer_wXZ9 · 2025-08-03
> > **Most Questions Addressed**
> >
> > Thank you for your detailed response and the comprehensive quantitative updated results. Most of my questions and concerns are addressed. However, I still have a minor question regarding the interpretation of the 3D exposure modeling.
> >
> > In the rebuttal, the authors claim that the 3D exposure learns an optimal exposure value per 3D point, which is still questionable in my opinion. As I mentioned previously, I think the "exposure" itself is a camera-related property rather than a spatial property, so I don't understand what the "optimal exposure" means as a spatial property. I think the only spatial property associated with the 3D point should be its radiance, while the exposure controls per camera determine how to convert the spatial radiance to the pixel brightness.
> >
> > Nevertheless, since the rebuttal provides quantitative results showing that the 3D exposure modeling benefits the performance, I consider it a useful empirical design. I would be glad if the authors provided more physical interpretation of it during the rebuttal period.

---

> > > ### Author Response · Authors · 2025-08-04
> > >
> > > Thank you for your engagement and your interesting question regarding the physical interpretation of our 3D exposure modeling. We are glad that our rebuttal has addressed your major concerns and questions.
> > >
> > > We agree that, in the traditional sense, exposure is a camera-dependent property. However, our method's "optimal exposure" is a conceptual departure acting as a spatially-varying variable. It is not directly a physical property but rather a learned representation that guides our model to produce high-quality, well-exposed images, similar to how a photographer might locally adjust exposure in high dynamic range photography.
> > >
> > > In high dynamic range scenes, a single camera exposure setting cannot capture all parts of the scene optimally. By learning a 3D exposure field, our model finds a per-point exposure value that prevents clipping and under or over-saturation for each point. When we render a novel view, our system uses this field to generate a 3D-consistent tonemap, resulting in a robust, high-quality, and perceptually pleasing image that further produces consistent colors across the entire scene (see e.g. the fly-through videos from the supp. mat.).
> > >
> > > In summary, our approach uses an empirically-derived, spatially-varying "optimal exposure" to handle diverse lighting and to produce 3D consistent, well-exposed colors in all views. We will add this additional explanation to the final version of our manuscript. Thank you once again for your constructive feedback.

---

> > > > ### Comment · Reviewer_wXZ9 · 2025-08-05
> > > >
> > > > Thanks for your explanation and clarification! I think your explanation of “spatially varying learnable representation rather than a physical property” makes sense, and I look forward to seeing you updating such discussions in the revised paper. I don’t have further concerns and I will raise my score to 5.

---

> ### Comment · Area_Chair_PuCj · 2025-08-03
> **Acknowledge Authors' Response**
>
> Dear Reviewer wXZ9,
>
> The authors have provided responses to your questions. What is your view after seeing this additional information? It would be good if you could actively engage in discussions with the authors during the discussion phase ASAP, which ends on EoA (Aug 6).
>
> Best,
> AC

---

### Official Review · Reviewer_Lgpm · 2025-06-15

**Clarity:** 3
**Significance:** 3
**Originality:** 2
**Rating:** 5
**Confidence:** 3

**Summary:**

The paper tackles the problem of learning a NeRF representation in scenes where input images suffer from severe inconsistencies between captures (e.g., exposure differences), which can break multiview consistency and cause artifacts. To address this, the authors propose learning a 3D exposure field defined per 3D point rather than per image, as done in previous works. This approach enables exposure to be optimized jointly with the scene representation, resulting in 3D-consistent and well-exposed renderings without requiring post-processing or reference views. The method achieves state-of-the-art quantitative results by a large marginand is also the fastest to train, with a 3× speedup over the second fastest method.

**Questions:**

The 3D exposure field is a core contribution of the paper and is designed to improve consistency and smoothness across viewpoints—qualities that are best appreciated through video sequences rather than per-frame metrics. The included videos already show high-quality results, but I believe the paper would benefit from more targeted qualitative comparisons to help isolate the specific impact of the 3D exposure field.

For instance, a side-by-side video comparison between:

-the proposed 3D exposure field,
-and a static embedding from test/train images

could highlight how much the 3D modeling contributes to temporal coherence, scene clarity, and exposure stability—especially in challenging HDR scenes.

**Ethical Concerns:**

["NO or VERY MINOR ethics concerns only"]

**Final Justification:**

I like the paper and I have no further questions, I confirm my rating as 5.

**Limitations:**

yes

**Paper Formatting Concerns:**

All good

**Quality:**

3

**Strengths And Weaknesses:**

Strenghts

-State-of-the-art results and speed: The proposed method achieves the best quantitative results across multiple benchmarks and is also the fastest to train—3× faster than the second-best approach.

-Simple but effective ideas: The core contribution (a 3D neural exposure field) is conceptually simple yet elegant and leads to meaningful improvements.

-Well-justified regularization: The introduction of well-exposedness, saturation, and smoothness regularization terms is intuitive. These terms help the model infer more realistic exposure levels per 3D point, effectively guiding the learning of correct appearance across varying exposure conditions. Ablation studies confirm their impact.

-Weaknesses
It is unclear why the authors compare to original NeRF rather than ZipNeRF in Table 1, especially since ZipNeRF is their actual architectural backbone.

---

> ### Author Rebuttal · Authors · 2025-07-30
>
> Thank you for your thorough review. We are encouraged by your description of our method as the one with "the best quantitative results across multiple benchmarks" while it "is also the fastest to train", and are delighted to hear that our core contributions are "simple but effective ideas". We address concerns in the following:
>
> > It is unclear why the authors compare to original NeRF rather than ZipNeRF in Table 1, especially since ZipNeRF is their actual architectural backbone.
>
> Thank you for your insightful comment. In Table 1, we refer to the original NeRF and report it as it is part of the common benchmark (see also [1] and [2]). However, we agree that this comparison would benefit from also reporting ZipNeRF which produces the following results:
>
> | Method | PSNR  | SSIM  | LPIPS  |
> |---|---|---|---|
> | NeRF  | 14.24 |  0.539   | 0.402  |
> | ZipNeRF  | 19.37  | 0.703  | 0.133     |
> | Ours  | 40.45 |  0.986 |  0.017  |
>
> We updated Table 1 of the main paper.
>
> [1] *Huang, Xin, et al. "Hdr-nerf: High dynamic range neural radiance fields." Proceedings of the IEEE/CVF Conference on Computer Vision and Pattern Recognition. 2022.*
>
> [2] *Cai, Yuanhao, et al. "Hdr-gs: Efficient high dynamic range novel view synthesis at 1000x speed via gaussian splatting." Advances in Neural Information Processing Systems 37 (2024): 68453-68471.*
>
> > The 3D exposure field is a core contribution of the paper and is designed to improve consistency and smoothness across viewpoints—qualities that are best appreciated through video sequences rather than per-frame metrics. The included videos already show high-quality results, but I believe the paper would benefit from more targeted qualitative comparisons to help isolate the specific impact of the 3D exposure field. For instance, a side-by-side video comparison between: -the proposed 3D exposure field, -and a static embedding from test/train images could highlight how much the 3D modeling contributes to temporal coherence, scene clarity, and exposure stability—especially in challenging HDR scenes.
>
> Thank you for your time to analyse not only the paper but also the supp. mat. which includes the video fly-throughs you are referencing. We find your comment encouraging and agree that such a qualitative comparison would be helpful for the reader. Unfortunately, we are not allowed by official NeurIPS guidelines to attach any image or video material to the rebuttal, but we will include this in the final version and a project page.

---

> ### Comment · Area_Chair_PuCj · 2025-08-03
> **Acknowledge Authors' Response**
>
> Dear Reviewer Lgpm,
>
> The authors have provided responses to your questions. What is your view after seeing this additional information? It would be good if you could actively engage in discussions with the authors during the discussion phase ASAP, which ends on EoA (Aug 6).
>
> Best,
> AC

---

> > ### Author Response · Authors · 2025-08-07
> >
> > Dear Reviewer Lgpm,
> >
> > We hope that we addressed your concerns and questions in our rebuttal. Please let us know if you need additional clarification or information.
> >
> > Thanks!

---

### Official Review · Reviewer_oJDR · 2025-06-26

**Clarity:** 3
**Significance:** 3
**Originality:** 3
**Rating:** 4
**Confidence:** 3

**Summary:**

This work introduces neural exposure fields to model the 3D exposure, which optimizes the exposure field and scene representation together. It bypasses the need for post-processing steps or multi-exposure captures. Experimental results verify its effectiveness on challenging scenes.

**Questions:**

Please review the "Weakness".

**Ethical Concerns:**

["NO or VERY MINOR ethics concerns only"]

**Final Justification:**

My main concern in the previous stage was that the exposure field depends on both the 3D position and view direction. The authors have provided further analysis to address this point. Overall, the paper presents a solid idea, comprehensive experiments, and clear writing and analysis. I therefore recommend acceptance.

**Limitations:**

Yes

**Quality:**

2

**Strengths And Weaknesses:**

Strengths:
- The paper is well-written and easy to follow.
- The learning of the exposure field results in strong rendering performance, demonstrating improved performance on challenging scenes.

Weakness:
- It is not entirely convincing that the exposure field depends solely on the 3D point. In my understanding, exposure is also partially influenced by the viewing direction. The exposure conditioning approach proposed in Fig. 5b may therefore not be the most appropriate design. Although the method achieves better performance compared to Fig. 5a, the improvement appears to be more of an engineering modification rather than a fundamentally grounded design choice.
- There is no parameter and time comparison. As seen in Fig. 5, the proposed method is built on HDR-NeRF, so it might be convincing to also compare the parameters and efficiency with HDR-NeRF.
- It is recommended to replace the proposed Latent Exposure Conditioning in Fig. 5b by Fig. 5a in the ablation study, making it more straightforward to demonstrate the effectiveness of the proposed method.

---

> ### Author Rebuttal · Authors · 2025-07-30
>
> Thank you for the thoughtful feedback on our submission. Describing our paper as "well-written" and our method as having "strong rendering performance" on "challenging scenes" is motivating for us. We address your concerns below:
>
> > It is not entirely convincing that the exposure field depends solely on the 3D point. In my understanding, exposure is also partially influenced by the viewing direction.
>
> Thanks for your comment. We agree that exposure itself is view-dependent as it is a scalar that is defined per image and hence camera pose. In this work, our goal is not to reproduce input exposure, but instead learn an **optimal exposure value** per 3D point. This enables, as can be seen in the fly-through videos in the supp. mat., smooth and stable color prediction across the entire scene, while per image/camera pose exposures can vary drastically as shown in Figure 2 of the main paper.
> Further, below we report an additional ablation study comparing our default method to our method with the modification of adding the viewing direction onto the bottleneck vector, producing similar results:
>
> | Method | PSNR  | SSIM  | LPIPS  |
> |---|---|---|---|
> | Ours | 41.25 | 0.993   | 0.008  |
> | Ours + View. Dep.  | 41.33 | 0.993  | 0.008  |
>
>  > The exposure conditioning approach proposed in Fig. 5b may therefore not be the most appropriate design. Although the method achieves better performance compared to Fig. 5a, the improvement appears to be more of an engineering modification rather than a fundamentally grounded design choice.
>
> We appreciate the comment and agree that operating in a latent space instead of on RGB color directly adds a layer of abstraction. This design choice leads to a significant performance improvement of 45% in MSE (+2.7dB in PSNR), as can be seen in Table 2 of the main paper and below in the final answer. We therefore believe that it is still a valuable contribution and design choice. Further, we note that operating in latent space is common in modern computer vision models and has led to breakthroughs (see e.g. latent diffusion in generative modeling (LDMs) or positional encoding in view synthesis (NeRF)).
>
> > There is no parameter and time comparison. As seen in Fig. 5, the proposed method is built on HDR-NeRF, so it might be convincing to also compare the parameters and efficiency with HDR-NeRF.
>
> In the following, we report number of parameters and render time on the HDRNeRF dataset next to the training time (that is also reported in Table 1 of the main paper) for HDR-NeRF and our method:
>
> | Method | Train Time (min)  | Params (mio)  | Render Time (sec)  |
> |---|---|---|---|
> | HDR-NeRF  | 542 | 1.19   | 3.39  |
> | Ours  | 11 | 154.62  | 0.07     |
>
> We find that our model trains ~49x faster and renders ~48x faster. We achieve this large training and test time speed up by incorporating a multi-resolution hash grid backbone (see also [1] and [2]). Compared to HDR-NeRF’s MLP architecture, this increases the number of parameters by two orders of magnitude. In most practical use cases, training and in particular test time speed is significantly more important than the parameter count which can also be seen by the large adoption of grid-based architectures since the introduction in InstantNGP [1].
>
> *[1]: Müller, Thomas, et al. "Instant neural graphics primitives with a multiresolution hash encoding." ACM transactions on graphics (TOG) 41.4 (2022): 1-15.*
>
> *[2]: Barron, Jonathan T., et al. "Zip-nerf: Anti-aliased grid-based neural radiance fields." Proceedings of the IEEE/CVF International Conference on Computer Vision. 2023.*
>
> > It is recommended to replace the proposed Latent Exposure Conditioning in Fig. 5b by Fig. 5a in the ablation study, making it more straightforward to demonstrate the effectiveness of the proposed method.
>
> In the ablation study in Table 2 of the main paper, we find that our default model improves over 45% in MSE compared to our model without the latent exposure conditioning  (see also answer 2).

---

> > ### Comment · Reviewer_oJDR · 2025-08-03
> > **Reply to authors**
> >
> > Thank the authors for their detailed responses. The results demonstrate that view-dependent 3D exposure modeling still holds significant potential, as evidenced by the higher PSNR (41.33).
> >
> > Besides, compared to Fig. 5a, the structure in Fig. 5b includes additional layers (including two MLPs), and deeper networks generally have stronger representation capabilities. Therefore, it remains difficult to isolate and verify the effectiveness of the proposed approach.

---

> > > ### Author Response · Authors · 2025-08-04
> > >
> > > Thank you for your time and your comments on the rebuttal. We are happy that our rebuttal was able to address most of your concerns and we address your remaining ones in the following.
> > >
> > > > The results demonstrate that view-dependent 3D exposure modeling still holds significant potential, as evidenced by the higher PSNR (41.33).
> > >
> > > Thanks. Please note that in the above-shared ablation study, both **SSIM and LPIPS are the same** with a precision of three digits after comma (0.993 and 0.008), and the **PSNR changes by 0.08** (from 41.25 to 41.33) which is usually not considered significant. However, we agree with you that this additional ablation and discussion is insightful so that we will add it to the final version.
> > >
> > > > Besides, compared to Fig. 5a, the structure in Fig. 5b includes additional layers (including two MLPs), and deeper networks generally have stronger representation capabilities. Therefore, it remains difficult to isolate and verify the effectiveness of the proposed approach.
> > >
> > > Please note that in Figure 5, as described in the caption, we compare the MLP architecture and exposure conditioning of HDRNeRF [1] with our (shallower) MLP architecture and exposure conditioning. In the above-discussed ablation study in Table 2 of our main paper, we compare our default method (with latent exposure conditioning) to our method with non-latent exposure conditioning ("w/o Lat. Con.") where the only architectural difference is **where** the exposure is injected while no additional layers or MLPs are used. We are therefore convinced that this ablation study is valid and shows the significant stronger performance of our default method (+45% in MSE). We will add these details and discussion to the main paper to make this clearer, thanks for pointing this out!
> > >
> > >
> > > [1] *Xin Huang, Qi Zhang, Ying Feng, Hongdong Li, Xuan Wang, and Qing Wang. HDR-NeRF: High Dynamic343Range Neural Radiance Fields. IEEE Conf. Comput. Vis. Pattern Recog., 2022.*

---

> > > > ### Comment · Reviewer_oJDR · 2025-08-06
> > > > **Reply to authors**
> > > >
> > > > Thank you for the reply, which has addressed most of my concerns. I will raise my rating.

---

> ### Comment · Area_Chair_PuCj · 2025-08-03
> **Acknowledge Authors' Response**
>
> Dear Reviewer oJDR,
>
> The authors have provided responses to your questions. What is your view after seeing this additional information? It would be good if you could actively engage in discussions with the authors during the discussion phase ASAP, which ends on EoA (Aug 6).
>
> Best,
> AC

---

### Official Review · Reviewer_SEiX · 2025-07-05

**Clarity:** 3
**Significance:** 3
**Originality:** 3
**Rating:** 4
**Confidence:** 3

**Summary:**

This paper introduces Neural Exposure Fields (NExF), a method for view synthesis that learns optimal exposure values directly in 3D space to handle scenes with strong lighting variations. Instead of relying on per-image exposure or 2D tonemapping, NExF predicts exposure per 3D point and integrates it into the training of a neural radiance field using a novel latent exposure conditioning mechanism. The method emphasizes well-exposedness and saturation in its loss design and uses regularization to enforce spatial smoothness, leading to high-quality, consistent reconstructions. NExF outperforms prior work in both accuracy and efficiency, achieving over 55% improvement in MSE while requiring less training time and no HDR supervision. It demonstrates strong performance even on complex real-world and in-the-wild datasets, producing photorealistic results without manual tuning or reference views.

**Questions:**

It would be valuable to evaluate the method on temporally ordered sequences and discuss whether temporal smoothness constraints or recurrent modeling might improve consistency in dynamic or video-based applications.

The training process assumes access to per-image exposure values. How sensitive is the method to noise or inaccuracy in this metadata?

It may be helpful to explore adaptive or learned weighting schemes, or to analyze performance under extreme lighting variations to evaluate how generalizable the method is beyond the studied datasets.

**Ethical Concerns:**

["NO or VERY MINOR ethics concerns only"]

**Final Justification:**

Thanks for your explanation and clarification in rebuttal, which resolves most of my concerns.
I keep my original rating of borderline accept.

**Limitations:**

The limitations are briefly discussed.

**Quality:**

2

**Strengths And Weaknesses:**

Strengths: The paper introduces a novel idea of learning exposure values directly in 3D space, rather than per image or per pixel. This leads to exposure-aware radiance fields that produce consistent and well-exposed renderings even in scenes with high dynamic range, such as indoor-outdoor transitions. The method outperforms existing baselines in both accuracy and training efficiency, achieving clear MSE improvement while requiring no HDR images or manual reference views.


Weaknesses:
1. The exposure field is learned independently of time or sequence, so for video or sequential input, it’s unclear whether predictions would be temporally stable.
2. The training process assumes access to accurate per-image exposure metadata, which may not always be available.
3. The well-exposedness and saturation weights are based on fixed formulas and hyperparameters. Although empirically effective. How about the sensitivity of the method to hand-crafted terms and the generalization in drastically different lighting distributions?

---

> ### Author Rebuttal · Authors · 2025-07-30
>
> Thank you for your time and careful review of our work. We find it encouraging that you describe our work as "introducing a novel idea" that "outperforms baselines in both accuracy and training efficiency". In the following, we address your concerns:
>
> > Weakness: The exposure field is learned independently of time or sequence, so for video or sequential input, it’s unclear whether predictions would be temporally stable.
>
> It is correct that we address the standard view synthesis task where the input capture is not assumed to be sequential. It is important to note that this is the more general assumption, hence our method can also be used for sequential input. Thanks to our 3D representation of both 3D geometry/appearance and exposure, we find that rendered videos showing fly throughs of our reconstructed scenes appear smooth and stable (please see the videos attached in the supp. mat.). Further, in Figure 2 of the main paper, we show a qualitative comparison of our method with traditional 2D tonemapping of two close-by frames. While our method produces stable colors for the same 3D point, the 2D tonemapping operation leads to significant color changes for the same 3D point despite the close proximity of the camera viewpoints. We conclude that even dense sequential input can exhibit significant color changes while our method - thanks to operating in 3D - leads to a smooth and stable color prediction.
>
> > Corresponding Question: It would be valuable to evaluate the method on temporally ordered sequences and discuss whether temporal smoothness constraints or recurrent modeling might improve consistency in dynamic or video-based applications.
>
> For the first part, please see the answer for the first weakness above. In addition, we agree that dynamic 3D reconstruction / 4D reconstruction would be an interesting avenue for our proposed system. Given that this would require significant additional research going beyond the proposed system, we identify this as promising future work we aim to address soon.
>
> > Weakness: The training process assumes access to accurate per-image exposure metadata, which may not always be available.
>
> > Corresponding Question: The training process assumes access to per-image exposure values. How sensitive is the method to noise or inaccuracy in this metadata?
>
> Thanks for the interesting question. Indeed, similar to related works, we assume to have per-image exposure given during training. To quantify the robustness of our method toward noise in the input exposure, we ran an ablation study on the HDRNeRF Dog scene. We augment the input exposure with Gaussian noise with different standard deviation levels varying from 1e-5 to 1e-2 (while exposure is normalized to be in the range (0, 1]):
>
> | Noise | PSNR  | SSIM  | LPIPS  |
> |---|---|---|---|
> | 0.0  | 41.25 | 0.993   | 0.008  |
> | 1e-5  | 41.17  | 0.993  | 0.009     |
> | 1e-4  | 40.71 |  0.992 |  0.011  |
> | 1e-3  | 36.98 |  0.976 |  0.029  |
> | 1e-2 | 15.70 |  0.513 |  0.537  |
>
> We observe that our model is robust toward noise levels with a standard deviation of up to 1e-4. For a noise level of 1e-3, results slightly worsen while still exhibiting high PSNR/SSIM/LPIPS metrics. For a standard deviation of 1e-2, the input exposure is too drastically corrupted leading to drop in performance. We added this to the final manuscript.
>
> >  Weakness: The well-exposedness and saturation weights are based on fixed formulas and hyperparameters. Although empirically effective. How about the sensitivity of the method to hand-crafted terms and the generalization in drastically different lighting distributions?
>
> > Corresponding Question: It may be helpful to explore adaptive or learned weighting schemes, or to analyze performance under extreme lighting variations to evaluate how generalizable the method is beyond the studied datasets.
>
> Thank you for your suggestion. The current evaluations already contain drastic lighting variations (see e.g. Figure 2 of the supp. mat.), but we agree that evaluating even more extreme / out-of-distribution (OOD) lighting conditions are interesting to explore. To this end, we trained our method on the HDRNeRF Dog scene where input training images were only sampled with exposure values t_2 and t_4, and we report evaluation on the test set images for all exposure values t_1 - t_5, where t_1 < t_2 < t_3 < t_4 < t_5 (see also Section 4.1 of the main paper). Different from the evaluation in Table 1 of the main paper, the exposures t_1 and t_5 are not only not observed during training but even outside of the observed exposure range [t_2, t_4] which depicts an even more extreme case of OOD data.
>
> | Exposure Set | PSNR  | SSIM  | LPIPS  |
> |---|---|---|---|
> | ID: {t_2, t_4}  | 41.62 | 0.995   | 0.006  |
> | OOD: {t_3}  | 36.17  | 0.987  | 0.018     |
> | Extreme OOD: {t_1, t_5}  | 30.83 |  0.977 |  0.033  |
>
> We find that our model produces high-quality images for all sets, including  the extreme out-of-distribution (OOD) scenario. Quantitatively, this can be seen in particular in the high SSIM and LPIPS metrics for all exposure sets. The drop in PSNR, in particular for the extreme OOD scenario, can be explained by the fact that the PSNR metric is directly comparing per-pixel RGB values while our model never observes any reference image for such exposure values as they are even outside of the training range. We added this to the final manuscript.

---

> > ### Comment · Reviewer_SEiX · 2025-08-06
> > **Response to rebuttal**
> >
> > Thanks for your explanation and clarification! I think the rebuttal resolves most of my concerns. I don't have more questions to raise.

---

> ### Comment · Area_Chair_PuCj · 2025-08-03
> **Acknowledge Authors' Response**
>
> Dear Reviewer SEiX,
>
> The authors have provided responses to your questions. What is your view after seeing this additional information? It would be good if you could actively engage in discussions with the authors during the discussion phase ASAP, which ends on EoA (Aug 6).
>
> Best,
> AC

---

### Comment · Area_Chair_PuCj · 2025-08-02
**Discussion with Authors**

Dear Reviewers,

The discussion period with the authors has now started. It will last until Aug 6th AoE. The authors have provided responses to your questions. I request that you please read the authors' responses, acknowledge that you have read them and start discussions with the authors RIGHT AWAY if you have further questions, to ensure that the authors enough time to respond to you during the discussion period.

Best,
AC

---

### Note · Authors · 2025-08-11

Dear Area Chair and Reviewers,

Thank you for the highly constructive and thorough review process. The discussions have been of high value and have led to significant improvements that will be incorporated into the final submission.

Specifically, based on your feedback, we conducted and will add:
- New ablation studies on robustness to noisy metadata and out-of-distribution generalization.
- Additional baseline comparisons, including direct quantitative comparisons to ZipNeRF (on the HDRNeRF dataset) and Bilarf.
- A detailed efficiency and parameter comparison with HDR-NeRF, demonstrating our method is ~49x faster in training and ~48x faster in rendering.
- A clarified description of our core motivation behind our 3D exposure field and latent conditioning mechanism, framing it as a spatially varying learnable representation for achieving 3D consistency rather than a direct physical property.

We are pleased that our responses resolved the initial concerns. Reviewer Lgpm, who was already positive in their initial review with a score of 5, acknowledged our response. In addition, Reviewer SEIX noted their questions were answered, and both Reviewer oJDR and Reviewer wXZ9 explicitly stated they would raise their ratings, with Reviewer wXZ9 raising their score to 5.

We are confident that the strengthened paper is a solid contribution. Thank you again for your engagement and guidance. We hope for your positive consideration.

---

### Decision · Program_Chairs · 2025-09-17

**Decision:**

Accept (poster)

**Comment:**

This paper proposes a novel method to jointly estimate both a scene's representation and its exposure enabling reconstruction of high dynamic range scenes captured in the wild. It received final ratings of 2 x borderline accept and 2 x accept from four expert reviewers. The reviewers appreciated the work for its novelty and strong quantitative performance. The reviewers concerns were sufficiently addressed during the reviewer-author discussion phase. The AC concurs with the reviewers' consensus and recommends acceptance. Congratulations! The authors should include the changes that they have promised in their final camera ready paper.